# Laparoscopic Sleeve Gastrectomy versus Laparoscopic Roux-en-Y Gastric Bypass: An Analysis of Weight Loss Using a Multilevel Mixed-Effects Linear Model

**DOI:** 10.3390/jcm12062132

**Published:** 2023-03-08

**Authors:** Camille Pouchucq, Olivier Dejardin, Véronique Bouvier, Adrien Lee Bion, Véronique Savey, Guy Launoy, Benjamin Menahem, Arnaud Alves

**Affiliations:** 1Department of Digestive Surgery, University Hospital of Caen, 14032 Caen, France; 2UMR INSERM 1086 ‘ANTICIPE,’ 14076 Caen, France; 3Department of Research, University Hospital of Caen, 14033 Caen, France; 4Registre des Tumeurs Digestives du Calvados, 14076 Caen, France

**Keywords:** bariatric surgery, weight loss, multilevel models

## Abstract

Background: Regarding weight loss outcomes, the results published after laparoscopic sleeve gastrectomy (LSG) and laparoscopic Roux-en-Y (LRYGB) are conflicting. At this time, no clear evidence exists that outcomes from LSG are similar to those for LRYGB. The main objective of this study was to compare the percent of total weight loss (%TWL) between LRYGB and LSG over the first 2 years using a multilevel mixed-effects linear regression. Methods: Data were collected from a prospectively maintained database of patients who underwent primary laparoscopic bariatric surgery from January 2016 to December 2017 at a French accredited bariatric center. The medical records of 435 consecutive patients were analyzed. %TWL was calculated at each follow-up surgical consultation and used as a repeated outcome variable in our models to assess the long-term %TWL. Due to this hierarchical structure of the data (%TWL at each visit = level 1) within patients (level 2), a multilevel linear regression adjusted for age, sex, preoperative BMI and comorbidities was used. Results: Among the medical records of 435 consecutive patients included, 266 patients underwent LRYGB and 169 underwent LSG. The average %TWL at 2 years was 31.7% for the LRYGB group and 25.8% for the LSG group. The final multivariate model showed that, compared with LRYGB, LSG was associated with a decreased %TWL at over 2 years of follow-up (β: −4.01; CI95%: −5.47 à −2.54; *p* ≤ 0.001). Conclusion: This observational study suggests that compared with LRYGB, LSG was associated with a decreased %TWL at 2 years using a multilevel model. Further studies are required to confirm the results observed with this statistical model.

## 1. Introduction

The last 30 years have brought a dramatic increase in the worldwide prevalence of obesity, causing considerable social and economic burden [1,2,3]. There is strong evidence that bariatric surgery, compared with medical treatment alone, results in substantial and long-term weight loss in morbidly obese patients [4,5,6]. Laparoscopic sleeve gastrectomy (LSG) and laparoscopic Roux-en-Y gastric bypass (LRYGB) are currently the two most frequently performed procedures in bariatric surgery, and weight loss is considered to be the primary indicator of its success [7,8]. At least 70 randomized controlled trials (RCTs) or observational studies (OS) have directly compared LRYGB and LSG [9,10,11,12,13,14,15,16]. While the majority of RCTs have shown that patients undergoing LRYGB and LSG experience similar weight loss [13,14,16,17], OS have shown that LRYGB patients achieve greater weight loss than LSG patients [9,11]. Furthermore, in daily practice, the choice of procedure is based on a shared decision-making process [18] (i.e., patients’ own preferences and values, comorbidities, intraoperative findings, and surgeons’ expertise and habits) that may not be measured in RCTs; hence, there is interest in complementary observational studies. Finally, traditional approaches to assess long-term weight loss in bariatric surgery have relied on a single time-point analysis, either through bivariate or regression models. However, this analysis requires, from a statistical point of view, to take into account the presence of repeated weight measurements for the same individual over time. To preserve this hierarchical structure, multilevel mixed-effects linear models should be used to assess weight loss over time [19].

To our knowledge, no superiority in terms of long-term weight loss efficacy has been demonstrated between LSG and LRYGB, so there is no consensus.

The main objective was to compare the %TWL between LRYGB and LSG over the first 2 years using a multilevel mixed-effects linear regression.

## 2. Materials and Methods

### 2.1. Study Design

The medical records of 435 consecutive patients who underwent laparoscopic bariatric surgery between January 2016 to December 2017 were analyzed. The data were recorded prospectively. The inclusion criteria for this survey specified patients older than 18 years who underwent, at our specialized and accredited bariatric center, either LRYGB or LSG as a primary operation and had one or more weight measurements obtained during postoperative visits. All indications for bariatric surgery were assessed according to the International Federation for the Surgery of Obesity and Metabolic Disorders criteria [7,8] and endorsed in an interdisciplinary consensus meeting.

### 2.2. Surgical Technique

All procedures were standardized to be performed in the same way at our center. “The decision to perform LSG or RYGB took into account both the patient’s choice, maximum preoperative BMI and related comorbidities as well as the proposal of the multidisciplinary consultation meeting”. The surgical techniques used for this study were as previously described in the literature [20,21,22].

### 2.3. Data Collection

All relevant data included patient features (sex, age and comorbidities), pre- (surgery weight) and postoperative biometric values [weight, height, body mass index (BMI) and percentage of total weight loss (%TWL)]. The %TWL was calculated according to the following formula: [(surgery weight-follow-up weight)/surgery weight] × 100. %TWL was calculated at each follow-up surgical consultation and used as a repeated outcome variable in our models to assess the long-term %TWL.

All patients were assessed as part of a routine surgical follow-up program in the outpatient clinic and were seen on a regular schedule at approximately 1, 3, 6, 12, 18 and 24 months postoperatively. The exact day of each consultation was retrospectively collected for all patients. Thus, a delay corresponding to the difference between the day of the surgery and the day of the consultation was calculated.

### 2.4. Statistical Analyses

Chi-square and Fisher’s exact tests were used to identify statistically significant differences for descriptive comparisons between both groups. *p* < 0.05 was defined as statistically significant.

Efficacy in terms of weight loss was assessed using %TWL. The %TWL readings at each postoperative follow-up consultation for the same patient corresponded to the use of repeated data. Due to this hierarchical structure of the data (%TWL at each visit = level 1) within patients (level 2, n = 435), a multilevel mixed-effects linear regression adjusted for age, sex, preoperative BMI and comorbidities was used.

Subsequently, the continuous variable %TWL was introduced into the model as the most flexible form of a three-node restricted cubic spline (Mkspline in STATA).

*p* = 0.05 was considered significant in the final model. All statistical analyses were performed with Stata/SE version 13 (StataCorp, College Station, TX, USA).

This study was approved by the local medical ethics committee and was declared to the CNIL (2204611v0). The requirement for patient consent was waived owing to the retrospective nature of the study.

## 3. Results

### 3.1. Demographic and Clinical Characteristics

Between January 2016 and December 2017, 266 patients underwent LRYGB and 169 underwent LSG. Both groups were similar regarding age and obesity-related comorbidities (Table 1). The LRYGB group had a significantly higher prevalence of females. Conversely, the LSG group was significantly associated with both higher preoperative biometric values and ASA score. Both groups were similar regarding age and obesity-related comorbidities.

### 3.2. Weight Loss

The average %TWL at 2 years was 31.7% for the LRYGB group and 25.8% for the LSG group.

Negative β coefficients indicate lower long-term %TWL. Schematically, a significant variable with a negative β means that the patient lost less weight.

#### 3.2.1. Two Years %TWL

The final multivariate model shows that compared with LRYGB, LSG was associated with a decreased %TWL at over 2 years of follow-up (β: −4.01; IC 95%: −5.47 à −2.54; *p* ≤ 0.001) (Table 2). 

#### 3.2.2. Weight Loss Curve

To test the hypothesis of linearity due to the inclusion of the %TWL in a continuous form, we used a four-node cubic spline model. Using a spline (Figure 1), we found that after the two procedures, the patients routinely experienced weight loss in a gradual fashion during the first 2 years. For each procedure, the mean %TWL peaked during the first year (approximately 16 months) after surgery and then fell within the normal distribution, similar to a bell curve.

## 4. Discussion

Using a multivariate linear regression in our study sample, we found that LSG compared with LRYGB was associated with a decrease in %TWL at 2 years of follow-up. These findings are compelling because the majority of current studies (including RCTs) have reported little or no difference in short-term weight loss between LRYGB and LSG [13,14,15,23,24,25,26,27,28,29]. A subset of other studies have found that LRYGB results in greater weight loss than LSG at 1 to 4 years of follow-up [9,10,12,30,31,32,33].

Patients of our survey with LSG were characterized by higher preoperative BMI as compared to those with RYGB, while baseline obesity-related comorbidities were similar. Importantly, even after adjusting for age, sex and preoperative biometric values, LSG remained associated with lower %TWL than LRYGB.

A number of RCTs have been devoted to compare weight loss outcomes between LSG and LRYGB, most of which have shown similar weight loss between the two techniques [13,14,34]. In a recent meta-analysis, including nine RCTs, Han and al. reported no significant difference in terms of %PEEP between the two procedures [−0.16 (95% CI: −0.52 to 0.19; *p* = 0.36)]. This conclusion is supported by two previous RCTs, with a follow-up over 5 years greater than 80% [13,14]. Even though they are randomized trials, these results should be interpreted with caution due to the limited number of patients included in each arm (between 20 and 100 patients) and the short follow-up time that has been traditionally reported [34,35,36].

Conversely, observational studies generally show greater weight loss in patients who received **RYGB** [9,11,12,37]. Thus, the multicenter study, PCORnet Cohort Study, compared the weight results in 32,208 patients who received **RYGB** against 29,693 patients who received SG [11]. The results show an average total weight loss at 5 years of 25.5% (95% CI, 25.1–25.9%) for **RYGB** versus 18.8% (95% CI, 18.0–19.6%) for **LSG**. That is a difference of 6.7% in terms of %PPT (95% CI, 5.8 to 7.7; *p* < 0.001) in favor of **RYGB**. Overall, the results of observational studies suggest that the difference observed between the two techniques is greater in a non-randomized setting and may be the consequence of unmeasured differences. Indeed, in daily practice, the choice of procedure is based on a process of shared decisions taking into account preoperatively the preferences of the patient, his comorbidities, the habits and the expertise of the surgeon and/or his center but also intraoperative findings [18]. It is therefore a set of essential factors **that** cannot be taken into account during an RCT; hence, the interest of complementary observational studies.

Today, one of the difficulties in synthesizing the literature on the subject is based on the lack of uniformity in the analysis tools used. Most experts agree that weight loss and gain should be expressed as a percentage of preoperative weight (which also has the advantage of being easy to use in clinical practice) [38,39]. However, primary studies have evaluated weight loss in other forms, in particular using the %PEP, the average loss of BMI, the weight lost or the percentage loss of excess BMI (%PE-BMI) [16].

Finally, traditional approaches to assess long-term weight loss have relied on a single time-point analysis, either through bivariate models or through regression models that do not respect the fundamental assumption of independence of observations [40]. Very few studies have applied statistical methods for the analysis of repeated measures [11,13,14]. The use of models that did not take this statistical specificity into account may have led to erroneous results.

Furthermore, several studies gave results about excess weight loss after bariatric surgery without multilevel analyses [13,14,41,42]. Thus, statistical analyses were performed with logistic regression model or variance analysis.

Although the data recording is done prospectively, it is a single-center retrospective study with all its inherent limitations. It is also an observational study, which risks unobserved confounding. Finally, the resolution of comorbidities could not be assessed due to a lack of available data, even though this is a parameter for determining the effectiveness of bariatric surgery. Our results must also be interpreted with caution due to the lack of data on long-term complications and quality of life, which are fundamental parameters to be taken into consideration. Similarly, the problem of loss to follow-up patients cannot be excluded, as reported in a recent review of the literature [43,44,45].

However, the study has also several strengths. Our cohort benefited from a relatively high number of patients (n = 435) and a low rate of loss to follow-up 2 years after surgery associated with a satisfactory 2-year follow-up rate (approximately 70%) compared to literature data [13,14,37,43]. This can be explained in particular by standardized patient monitoring within our team for several years; this is done both on a regular basis and over the long term. Indeed, the follow-up of patients has been standardized within our team for several years [20,21,22].

Finally, we applied a multilevel mixed-effects linear model. To our knowledge, only three studies previously considered recorded weights as repeated measures by applying an adapted statistical method with conflicting results. The SLEEVEPASS study was considered to have failed to meet the equivalence criteria set out in the study design, whereas LRYGB was significantly associated with greater excess weight loss at 5 years [13]. The SM-BOSS found no differences in the percentage of body mass index loss, and slight differences in the 5-year body mass index (approximately 1 kg/m^2^) were found between studies [14]. Conversely, the PCORnet Cohort Study, a large multicenter observational study, found that patients lost more weight with LRYGB than with LSG at 1, 3 and 5 years [36].

The main strengths of our study are based, in our opinion, on the application of a statistical model taking into account the presence of longitudinal data, on the use of recommended monitoring tools (%PPT in particular) [38,39] and on a comprehensive analysis of the literature. 

## 5. Conclusions

In this observational study reflecting daily practice, we found that the use of LSG compared with the use of LRYGB was associated with a decreased %TWL at 2 years using a multivariate linear regression. Taken together, the findings from observational studies suggest that the differences in weight loss outcomes between LRYGB and LSG are slightly larger in nonrandomized settings and may be due to unmeasured differences.

Further studies, either RCTs or OS, are needed to confirm the results observed with this statistical model, also taking into account long-term outcomes (i.e., the resolution of co-morbidities, the prevalence of long-term complications and patients’ quality of life).

## Figures and Tables

**Figure 1 jcm-12-02132-f001:**
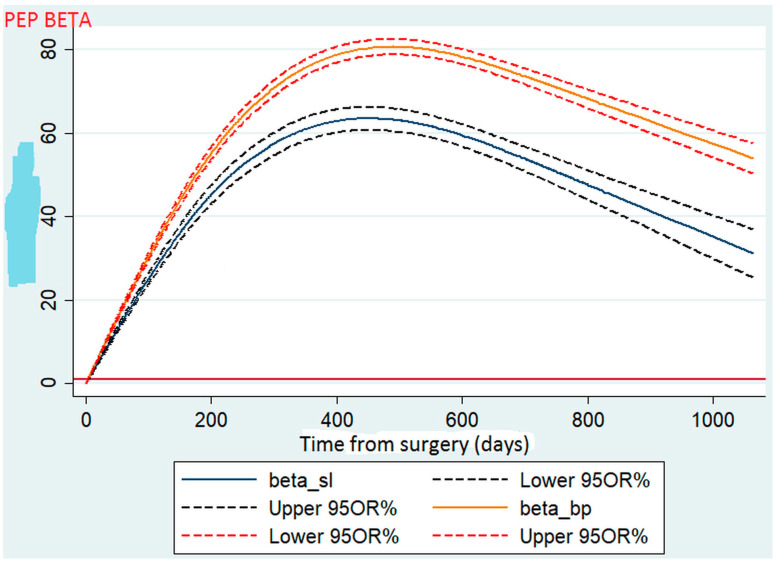
Comparison of total weight loss between sleeve gastrectomy and gastric bypass.

**Table 1 jcm-12-02132-t001:** Characteristics of patients (n = 435) who underwent bariatric surgery (LRYGB or LSG) at the University Hospital of Caen, 2016–2017.

	LRYGB		LSG		
	(n = 266)		(n = 169)		
Variables					*p* Values
**Sex**					**<0.001**
Female	227	85.3%	117	69.2%	
Male	39	14.7%	52	30.8%	
**Age**					0.382
Continuous (years), mean ± SD	266	43.4 ± 0.7	169	42.5 ± 0.9	
**Preoperative bodyweight**					**<0.001**
Continuous (kg), mean ± SD	266	110.1 ± 0.9	169	123.6 ± 1.8	
**Preoperative excess weight**					**<0.001**
Continuous (kg) mean ± SD	266	45.3 ± 0.7	169	56.1 ± 1.6	
**Preoperative BMI**					**<0.001**
Continuous (kg/m^2^) mean ± SD	266	40.5 ± 0.2	169	44.0 ± 0.6	
**ASA score**					**0.01**
2	222	83.4%	116	68.7%	
3	44	16.6%	53	31.3	
**Former smokers**					0.534
Yes	42	15.8%	23	13.6%	
No	224	84.2%	146	86.4%	
**Diabetes**					0.521
Yes	67	25.2%	38	22.5%	
No	199	74.8%	131	77.5%	
**Hypertension**					0.447
Yes	82	30.8%	58	34.3%	
No	184	69.2%	111	65.7%	
**Dyslipidaemia**					0.485
Yes	69	25.9%	49	29.0%	
No	197	74.1%	120	71.0%	
**Sleep apnoea**					0.184
Yes	118	44.4%	86	50.9%	
No	148	55.6%	83	49.1%	
**Length of hospital stay (days)**	266	3.4 ± 0.1	169	3.0 ± 0.1	**0.026**

**Table 2 jcm-12-02132-t002:** Linear mixed model of %TWL from 1 month to over 2 years of follow-up after bariatric surgery at the University Hospital of Caen, 2016–2017.

(n = 435)										
	n	Percentage	Univariable Analysis				Multivariable Analysis *			
Variables			β **	95% CI		*p* Values	β **	95% CI		*p* Values
**Sex**										
Female	1240	77.5%	Ref			**<0.001**	Ref			0.325
Male	359	22.5%	−5.46	−7.49	−3.42		−0.89	−2.65	0.88	
**Surgery age**										
Continuous	1599	100%	−0.29	−0.36	−0.22	**<0.001**	−0.15	−0.21	−0.09	**<0.001**
**Surgery BMI**										
Continuous	1599	100%	−1.3	−1.55	−1.05	<0.001	−1.17	−1.35	−0.98	<0.001
**Type of surgery**										
LRYGB	1069	66.8%	Ref			**<0.001**	Ref			**<0.001**
LSG	530	33.2%	−10.80	−12.56	−9.03		−6.67	−8.33	−5.00	
**Year of surgery**										
Continuous	1599	100%	−0.51	−0.81	−0.21	**<0.001**	0.48	0.21	0.74	**<0.001**
**Time since surgery**										
Continuous	1599	100%	0.78	0.75	0.81	**<0.001**	0.78	0.75	0.82	**<0.001**
**Former smokers**										
No	1368	85.5%	Ref			**0.017**	Ref			**0.003**
Yes	231	14.5%	2.97	0.52	5.41		3.14	1.06	5.20	
**Diabetes**										
No	1197	74.9%	Ref			**<0.001**	Ref			**0.003**
Yes	409	25.1%	−4.81	−6.73	−2.90		−2.63	−4.37	−0.88	
**Hypertension**										
No	1032	64.5%	Ref			**<0.001**				
Yes	567	35.5%	−6.15	−7.88	−4.41					
**Dyslipidaemia**										
No	1268	79.3%	Ref			**<0.001**				
Yes	331	20.7%	−4.51	−6.61	−2.40					
**Sleep apnoea**										
No	999	62.5%	Ref			**<0.001**				
Yes	600	37.5%	−6.59	−8.31	−4.87					
**Individual variance**										

* Multivariable final model after backward selection removing: dyslipidaemia (*p* = 0.815), hypertension (*p* = 0.630) and sleep apnoea (*p* = 0.375). ** Negative β coefficients indicate lower long-term percent excess weight loss.

## Data Availability

Not applicable.

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
