# Peer review of "Laparoscopic Sleeve Gastrectomy versus Laparoscopic Roux-en-Y Gastric Bypass: An Analysis of Weight Loss Using a Multilevel Mixed-Effects Linear Model"

_jcm, 2023, doi:10.3390/jcm12062132_

Round 1

Reviewer 1 Report

Pouchucq et al. describe %TWL evolution 2 years after LRYGB versus LSG in a single French bariatric center using a multilevel mixed-effects linear regression.  Furthermore, the results are compared to current literature.

They conclude that LSG was associated with a decreased %TWL ot 2 years using this method. They nicely explain that there is a dicrepancy between outcome of RCTs and OSs and thus the need for further research on this topic. 

The introduction, material and methods and results sections are concise and clear to read. However i have the following comments: 

- How many data points were captured? How were missing data handled?  

- How was the decision making proces for LSG versus LRYGB done? 

- If table 1 is correct, the LRYGB group should have a lower ASA score compared to LSG group.  

- Caption of figure 1 should be improved to make it more easy to understand 

- In Table 2 the line for surgery BMI is left blank?

The discussion section should be improved

- BPG? should be LRYGB i suppose? 

- % PEEP? % PPT? OS is used in stead of LSG (for BPG versus 18.8% (95% CI, 18.0%- 19.6%) for OS)

It is therefore a set of essential factors but cannot --> that in stead of "but"

either through bivariate models or through regression models that do not respect the fundamental assumption of independence of observations [ref].  --> reference is missing

- Furthermore, Several studies gave results about excess weigh loss after bariatric sur-gery without multilevel analyses. --> reference?

- sentences are repeated: 

A number of RCTs have been conducted comparing weight loss outcomes between LSG and LRYGB [13, 14, 26]. .... RCTs have been conducted to compare the weight outcomes of SG and BPG, mostly showing similar weight loss between the two techniques (16,17,39,40,65

- section is repeated: 

Page 6: In the SLEEVEPASS, the 5-year per-centage of excess weight loss was significantly higher with LRYGB; however, it was con-sidered not clinically significant since criteria of equivalence established in the design of the study were not fulfilled. The SM-BOSS found no differences in the percentage of body mass index loss, and slight differences in the 5-year body mass index (approximately 1 kg/m2) were found between studies. Conversely, the PCORnet Cohort Study, a large mul-ticenter observational study, found that patients lost more weight with LRYGB than with LSG at 1, 3, and 5 years.

Page 7:

In the SLEEVEPASS study, %PEP was higher after BPG at all follow-up times. However, being a non-inferiority study, this result was considered not clinically significant since the equivalence criteria established during the design of the study were not met. The SM-BOSS study showed no difference in terms of %PE-BMI and a slight difference in the change in BMI at 5 years (about 1 kg/m2). The large PCORnet Cohort Study (N=65,093) found that patients had a higher %PPT after BPG at 1, 3 and 5 years. Finally, in a very recent study, combining data from the SLEEVEPASS and SM-BOSS studies, Wölnerhanssen et al demonstrated the superiority of BPG in terms of %PE-BMI and %PPT.

Author Response

Reviewers’ comments

Reviewer 1 RW1

Pouchucq et al. describe %TWL evolution 2 years after LRYGB versus LSG in a single French bariatric center using a multilevel mixed-effects linear regression.  Furthermore, the results are compared to current literature. They conclude that LSG was associated with a decreased %TWL ot 2 years using this method. They nicely explain that there is a discrepancy between outcome of RCTs and OSs and thus the need for further research on this topic. 

The introduction, material and methods and results sections are concise and clear to read.

Authors’ response: Thank you for this comment which contributed to upgrade our academic work.

However i have the following comments: 

- How many data points were captured? How were missing data handled?  

Authors’ response: Thank you for this comment which contributed to upgrade our academic work.

We agree that this specific point could be improved. As explained in methods section, we used longitudinal data of weight loss using a multilevel linear regression. Thus, 435 patients (table 1) were follow during 2 years. At each follow-up surgical consultation (N=1599), the total weight loss was reported.

Concerning missing values, the total number of missing at each follow-up surgical consultation was respectively, all patients were seen at inclusion and 12 months after surgery. However, 36% of patients were lost of follow-up two year after surgery. since multiple imputations on hierarchical data is unusual, all analyses were performed in complete case analysis framework.

- How was the decision making process for LSG versus LRYGB done? 

Authors response : Thank you for this comment which contributed to upgrade our academic work.

Action in the manuscript : as requested by the reviewer The following sentence was added in the revised version of the manuscript  “The decision to perform LSG or RYGB took into account both the patient's choice, maximum preoperative BMI  and related comorbidities as well as the proposal of the multidisciplinary consultation meeting”.

- If table 1 is correct, the LRYGB group should have a lower ASA score compared to LSG group. 

Authors response : Thank you for this comment which contributed to upgrade our academic work.

Action in the manuscript : as requested by the reviewer, the comparison of the ASA score between the two groups has corrected  in the revised version of the manuscript. Conversely, the LSG group was significantly associated with both higher preoperative biometric values and ASA score. »

- Caption of figure 1 should be improved to make it more easy to understand 

Authors response : Thank you for this comment which contributed to upgrade our academic work.

Action in the manuscript : caption has been improved

- In Table 2 the line for surgery BMI is left blank?

Authors’ response: Thank you for this comment which contributed to upgrade our academic work.

Action in the manuscript : we completed surgery BMI in the table 2

The discussion section should be improved

- BPG? should be LRYGB i suppose? 

Authors’ response: Thank you for this comment which contributed to upgrade our academic work.

Action in the manuscript : as requested by the reviewer all “BPG” terms have been replaced by “RYGB” terms in the discussion of the revised manuscript 

- % PEEP? % PPT? OS is used in stead of LSG (for BPG versus 18.8% (95% CI, 18.0%- 19.6%) for OS)

Authors’ response: Thank you for this comment which contributed to upgrade our academic work.

Action in the manuscript : as requested by the reviewer the “OS” term has been replaced by “LSG” term in the discussion of the revised manuscript 

- It is therefore a set of essential factors but cannot --> that in stead of "but")

Authors’ response: Thank you for this comment which contributed to upgrade our academic work.

Action in the manuscript : as requested by the reviewer “but” has been replaced by “in stead of ” in the sentence of the revised manuscript 

- either through bivariate models or through regression models that do not respect the fundamental assumption of independence of observations [ref].  --> reference is missing

- Furthermore, Several studies gave results about excess weigh loss after bariatric sur-gery without multilevel analyses. --> reference?

 Authors’ response: Thank you for this comment which contributed to upgrade our academic work.

Action in the manuscript : referecnes were added (43 to 47)

- sentences are repeated: 

A number of RCTs have been conducted comparing weight loss outcomes between LSG and LRYGB [13, 14, 26]. .... RCTs have been conducted to compare the weight outcomes of SG and BPG, mostly showing similar weight loss between the two techniques (16,17,39,40,65

Authors’ response: Thank you for this comment which contributed to upgrade our academic work.

Action in the manuscript : as requested by the reviewer, both sentences have been modified to avoid repetition in the revised manuscript : “A number of RCTs have been devoted to compare weight loss outcomes between LSG and LRYGB most of which have shown similar weight loss between the two techniques (13,14,36) ».

- section is repeated: 

Page 6: In the SLEEVEPASS, the 5-year per-centage of excess weight loss was significantly higher with LRYGB; however, it was con-sidered not clinically significant since criteria of equivalence established in the design of the study were not fulfilled. The SM-BOSS found no differences in the percentage of body mass index loss, and slight differences in the 5-year body mass index (approximately 1 kg/m2) were found between studies. Conversely, the PCORnet Cohort Study, a large mul-ticenter observational study, found that patients lost more weight with LRYGB than with LSG at 1, 3, and 5 years.

Page 7:

In the SLEEVEPASS study, %PEP was higher after BPG at all follow-up times. However, being a non-inferiority study, this result was considered not clinically significant since the equivalence criteria established during the design of the study were not met. The SM-BOSS study showed no difference in terms of %PE-BMI and a slight difference in the change in BMI at 5 years (about 1 kg/m2). The large PCORnet Cohort Study (N=65,093) found that patients had a higher %PPT after BPG at 1, 3 and 5 years. Finally, in a very recent study, combining data from the SLEEVEPASS and SM-BOSS studies, Wölnerhanssen et al demonstrated the superiority of BPG in terms of %PE-BMI and %PPT.

Authors’ response: Thank you for this comment which contributed to upgrade our academic work.

Action in the manuscript : as requested by the reviewer, both paragraphs have been synthesised into one to avoid repetition in the revised manuscript :

“Finally, we applied a multilevel mixed-effects linear model. To our knowledge, only 3 studies previously considered recorded weights as repeated measures by applying an adapted statistical method with conflicting results. The SLEEVEPASS study was considered to have failed to meet the equivalence criteria set out in the study design, whereas LRYGB was significantly associated with greater excess weight loss at 5 years. The SM-BOSS found no differences in the percentage of body mass index loss, and slight differences in the 5-year body mass index (approximately 1 kg/m2) were found between studies. Conversely, the PCORnet Cohort Study, a large multicenter observational study, found that patients lost more weight with LRYGB than with LSG at 1, 3, and 5 years.”

Reviewer 2 Report

This is an interesting and well written paper. It is, however, a retrospective study with all the limitations that brings. My concerns are (1) the baseline BMI appears higher in the LSG group, so the two groups are not matched. Also, the discussion/conclusions are all about the weight loss. there is little or no discussion about the other relevant issues, ie complications, post-op quality of life, etc. The conclusions are justified but she be taken in this context and this should be discussed.

Author Response

Reviewers’ comments

Reviewer 2 : RW2.1.

This is an interesting and well written paper. It is, however, a retrospective study with all the limitations that brings. My concerns are (1) the baseline BMI appears higher in the LSG group, so the two groups are not matched. Also, the discussion/conclusions are all about the weight loss. there is little or no discussion about the other relevant issues, ie complications, post-op quality of life, etc. The conclusions are justified but she be taken in this context and this should be discussed.

Authors’ response: Thank you for this comment which contributed to upgrade improve the manuscript.

Actions in the manuscript: as requested by the reviewer, we discussed the inherent limitations of

the study related to the lack of data on long-term morbidity and quality of life. in the revised

manuscript “ Our results must also be interpreted with caution due to the lack of data on long-term

complications and quality of life, which are fundamental parameters to be taken into consideration.

Similarly, the problem of lost to follow-up patients cannot be excluded, as reported in a recent

review of the literature.

(Auge MDejardin MMenahem BLee Bion ASavey VLaunoy G Bouvier VAlves A. Analysis of the Lack of Follow-Up of Bariatric Surgery Patients: Experience of a Reference CenterJ Clin Med. 2022 Oct 26;11(21):6310. doi: 10.3390/jcm11216310. Auge M, Menahem B, Savey V, Lee Bion A, Alves A.Long-term complications after gastric bypass and sleeve gastrectomy: What information togive to patients and practitioners, and why? J Visc Surg. 2022 Aug;159(4):298-308. doi: 10.1016/j.jviscsurg.2022.02.004)

« Further studies, either RCTs or OS, are needed to confirm the results observed with this statistical model, also taking into account long-term outcomes (i.e., the resolution of co-morbidities, the prevalence of long-term complications and patients' quality of life) »

Round 2

Reviewer 2 Report

It reads much better now.